# Performance Assessment of an Ultra Low-Cost Inertial Measurement Unit for Ground Vehicle Navigation

**DOI:** 10.3390/s19183865

**Published:** 2019-09-07

**Authors:** Rodrigo Gonzalez, Paolo Dabove

**Affiliations:** 1GridTICs, National University of Technology, Various M5502AJE, Argentina; 2Department of Environmental, Land and Infrastructure Engineering (DIATI), Politecnico di Torino, 10129 Torino, Italy

**Keywords:** MEMS, inertial sensors, MPU-6000, low-cost, ground navigation

## Abstract

Nowadays, navigation systems are becoming common in the automotive industry due to advanced driver assistance systems and the development of autonomous vehicles. The MPU-6000 is a popular ultra low-cost Microelectromechanical Systems (MEMS) inertial measurement unit (IMU) used in several applications. Although this mass-market sensor is used extensively in a variety of fields, it has not caught the attention of the automotive industry. Moreover, a detailed performance analysis of this inertial sensor for ground navigation systems is not available in the previous literature. In this work, a deep examination of one MPU-6000 IMU as part of a low-cost navigation system for ground vehicles is provided. The steps to characterize the performance of the MPU-6000 are divided in two phases: static and kinematic analyses. Besides, an additional MEMS IMU of superior quality is also included in all experiments just for the purpose of comparison. After the static analysis, a kinematic test is conducted by generating a real urban trajectory registering an MPU-6000 IMU, the higher-grade MEMS IMU, and two GNSS receivers. The kinematic trajectory is divided in two parts, a normal trajectory with good satellites visibility and a second part where the Global Navigation Satellite System (GNSS) signal is forced to be lost. Evaluating the attitude and position inaccuracies from these two scenarios, it is concluded in this preliminary work that this mass-market IMU can be considered as a convenient inertial sensor for low-cost integrated navigation systems for applications that can tolerate a 3D position error of about 2 m and a heading angle error of about 3 °.

## 1. Introduction and Motivation

In recent years, ground navigation systems have become very popular in the automotive industry. The massive use of advanced driver assistance systems (ADAS) in modern vehicles, and the effort of developing driverless vehicles are pushing ground navigation systems to be essential in future cars in years to come. These systems are equipped with a high number of different sensors, starting from Global Navigation Satellite System (GNSS) receivers up to an inertial measurement unit (IMU), odometers, laser scanner and cameras, among others. In some cases, where both high accuracy and precision are needed, high-cost and high-productive instruments are employed, while for other applications it could be possible to consider low-cost or mass-market devices.

Generally, the previous literature about low-cost navigation and positioning systems for ground vehicles covers MicroElectroMechanical System (MEMS) IMUs whose costs are higher when compared to ultra low-cost IMUs (∼USD 10). For example, in [1], a VG440CA-200 IMU from ACEINNA (∼USD 4495) is used to present a new positioning methodology for ground vehicles. Another example is the work from [2], where an STIM300 IMU from Sensonor (∼USD 500) serves to test a forward velocity model based on non-holonomic constraints. Nevertheless, some researchers have started to pay attention to some ultra low-cost inertial sensors. In [3], a new vehicle attitude estimator is evaluated by using measurements from an L3GD20 IMU by ST Microelectronics (∼USD 10). Unfortunately, this work is strongly focused on simulated inertial data and real inertial data results are vaguely exposed for short periods, from 15 to 45 s only.

The MPU-6000 IMU from InvenSense Inc. (San Jose, CA, USA) [4] is an ultra low-cost MEMS solution used in several real-world applications. Its price is around USD 6 per unit. This particular IMU and its family members (MPU-6050, MPU-9250) can be found in different mass-market products, from wearables and robotics to Internet-of-Thing applications. In particular, the MPU-6000 is extensible used in the field of control systems for ground and aerial robotics, where light weight and low-power consumption are critical design constraints. It has been chosen to be part of some popular open hardware autopilots as Pixhawk [5] and Beaglebone Blue [6], and commercial autopilots as the Parrot Bebop [7] as well.

Although this mass-market IMU is extensively used in robotics, to the best knowledge of the authors, a detailed performance analysis of this inertial sensor as part of a ground vehicle navigation system is not available in the previous literature. Therefore, it is not clear if this particular IMU could effectively be used in ground vehicle navigation systems. Thus, the main contribution of this work is to provide a deep examination of the MPU-6000 IMU to verify if this sensor can be used as part of a low-cost navigation system for ground vehicles.

The steps to profile the behavior of the MPU-6000 are divided in two stages known as static and kinematic analyses. Besides, one additional MEMS IMU of superior quality is analyzed applying the same procedures just for the purpose of comparison. This IMU is an Ekinox-D by SGB Systems [8] whose cost is around USD 47,000. Firstly, an MPU-6000 IMU and an Ekinox-D IMU are logged during 24 h to subsequently apply the Allan variance technique on these data to identify and to quantify the sources of noise for each inertial sensor. Additionally, a normality test is run on each inertial sensor’s static data to verify if particular inertial measurements come from a Gaussian distribution.

In a second stage, both IMUs are mounted on a car along with two GNSS sensors, one U-blox M8T receiver [9] and the internal Ekinox-D GNSS receiver. Measurements from all sensors are registered while the vehicle is driven through the streets of Turin city (Italy). Two GNSS scenarios are considered, one with GNSS normal operation and another where GNSS is forcefully blocked. Finally, the performances of three possible integrated navigation systems are analyzed. These systems are composed by an inertial navigation system (INS) plus a GNSS receiver, known as INS/GNSS systems for short.

The rest of this paper is organized as follows. Section 2 shows results from the static analysis for the two IMUs under study. Section 3 details the real-world kinematics datasets used in this work and the results of post-processing IMUs and GNSS to evaluate three integrated navigation systems. Finally, in Section 4, concluding remarks are mentioned.

## 2. Static Analysis

An IMU manufacturer generally provides a datasheet specifying typical types of inaccuracies from a particular product, but usually this information is a generalization from a production batch. Moreover, the levels of some type of errors are not even provided. Commonly, a more detailed profile from a specific unit is needed in order to later use the level of these sensors’ noises to configure a Kalman filter, which will be part of an integrated navigation system (INS/GNSS system). As a consequence, estimations of position, velocity, and attitude will be more accurate.

The Allan variance (AV) is a well-known technique that is commonly used to identify and to quantify inertial sensors’ stochastic noises, as quantization noise, random walks errors, and bias instability, among others. In applying the AV technique, it is mandatory to only process data from static measurements [10]. Theoretical foundations about the AV can be found in the literature [10,11,12] and are beyond the scope of this paper.

An MPU-6000 IMU unit as part of a PixHawk autopilot [5] was mounted in a leveled non-magnetic plate and logged over 24 h to subsequently apply the Allan variance on these data. Moreover, one additional MEMS IMU of superior quality was simultaneously analyzed applying the same procedure just for comparing both IMUs’ performances. This IMU is an Ekinox-D by SGB Systems [8]. This MEMS IMU is considered as a tactical-grade IMU, with a bias in-run instability around 0.5${}^{\circ }$/h. The complete procedure took place at one of DIATI’s laboratories. The arrangement of inertial sensors for static logging is shown in Figure 1. The Ekinox IMU was logged at 250 Hz, which is the default operating frequency for this sensor. On the other hand, the MPU-6000 IMU was limited to work at 200 Hz to guarantee that MPU-6000 inertial measurements would be saved on the SD card without losing any data. Both IMUs were powered by batteries, thus they were not affected by electrical noise from the power supply.

Deterministic errors such as static bias (SB), which is the sensor mean at a levelled and static position, and standard deviation (σ) can be obtained from each sensor from static measurements. The latter is related to the level of background noise present in the measurements. Magnitudes of these two deterministic errors from the six sensors in each IMU are shown in Table 1. It is worth saying that this information is not provided by the MPU-6000 datasheet.

According to the IMU grade classification from ([13], Table 4.1, p. 113), which is based on values of static bias, Ekinox’s accelerometers are considered as intermediate grade sensors ($\left[10^{-3}\;,\;10^{-2}\right]$ m$/s^{2}$), and MPU-6000’s accelerometers are regarded as tactical grade sensors ($\left[10^{-2},10^{-1}\right]$ m$/s^{2}$). On the other hand, Ekinox’s gyroscopes are in the range of tactical grade sensors ($\left[5\times 10^{-6}\;,\;5\times 10^{-4}\right]$ rad/s), while MPU-6000’s gyroscopes are automotive grade sensors ($>5\times 10^{-4}$ rad/s). According to this classification, it is worth noting that MPU-6000 accelerometers show better performances than MPU-6000 gyroscopes. This situation is not optimal from the point of view of an INS/GNSS system, since attitude errors derived from biases couple proportionally to time squared into velocity, and time cubed into position [13] (Section 5.6.1).

Table 1 also exposes that Ekinox has lower values of static biases, as expected. Analysing the σ values from Table 1, it can be said that Ekinox’s accelerometers noises are one order of magnitude lower than the MPU-6000’s ones. On the contrary, MPU-6000’s gyroscopes have a level of noise one order of magnitude lower when compared to the Ekinox’s gyroscopes.

The AV analysis is carried out by using the tools provided by NaveGo [14,15,16], which is an open-source MATLAB/GNU Octave toolbox for processing integrated navigation systems and performing inertial sensors analysis. Table 2 exhibits the results of applying the AV procedure on the static measurements. Errors that come up after AV analysis are dynamic biases (DB), or bias instabilities, angle random walks and velocity random walks for gyroscopes and accelerometers, respectively (RW), and bias correlation times (CT). Table 2 also includes the RW values provided by the MPU-6000 datasheet as noise power spectral densities [4] (pp. 12–13) at the first column. These values are 0.005${}^{\circ }$$\;/s/\;\sqrt{\mathrm{Hz}}$ for gyroscopes and 400 $\mu g/\;\sqrt{\mathrm{Hz}}$ for accelerometers, which must be transformed to SI units for a straight comparison to NaveGo units.

Figure 2 and Figure 3 show the AV plots from accelerometers and gyroscopes respectively for both IMUs.

Analyzing the MPU-6000’s RW values from Table 2, it can be confirmed that MPU-6000 gyroscopes have one order of magnitude lower level of noise than the Ekinox gyroscopes. For the rest of the AV profile values, Ekinox shows a better performance. With respect to the MPU-6000’s RW values provided by the manufacturer, RW datasheet values for accelerometers and gyroscopes are close to the values found by applying the AV procedure.

### 2.1. Test of Normality

It is known that the Kalman filter is an optimal estimation tool for state–space linear systems whose internal states and output signals are corrupted with Gaussian noise. Alternatively, if the Gaussian assumption cannot be guaranteed, a particle filter can be considered as a suitable algorithm for this type of systems [17] (Ch. 7). Therefore, it is important to determine if a particular sensor follows a Gaussian distribution in order to choose the best estimation filtering technique to fuse information in the context of an integrated navigation system.

A test of normality is run on each inertial sensor from Ekinox and MPU-6000 IMUs based on the Anderson–Darling (AD) procedure [18]. It returns the test decision for the null hypothesis that the data is from a population with a normal distribution. The result *H* is 1 if the test rejects the null hypothesis at the 5% significance level, or 0 otherwise.

The AD normality test receives a real data vector and two additional values, the mean and the standard deviation inferred from this vector. Then, it verifies if an ideal Gaussian distribution built with those two statistical values matches the real probability distribution from the real data. Table 3 shows the results of running an AD test on every inertial sensor’s static data from MPU-6000 and Ekinox.

According to the results of the Anderson–Darling test from Table 3, gyroscopes from MPU-6000 do not follow a normal distribution according to the statistical values from Table 1, while the rest of the inertial sensors do. To verify this situation, the histogram and the cumulative distribution function (CDF) for each gyroscope are plotted, as shown in Figure 4. Additionally, Figure 5 exposes a graphical analysis of normal distribution for MPU-6000 accelerometers for completeness.

Figure 4 reveals that, although MPU-6000 gyroscopes do not follow a perfect Gaussian distribution, they can be modelled as pseudo-Gaussian processes. This situation will impact in the Kalman filter tuning (Section 3.3).

## 3. Kinematic Analysis

In the following section, the performance from an MPU-6000 IMU is analyzed in the context of a kinematic trajectory. Both kinematic and reference datasets are described in detail. The dynamic analysis will focus on three proposed stretches from the trajectory. Lastly, results from three proposed INS/GNSS systems are exposed and commented.

### 3.1. Kinematic Dataset Description

The kinematic dataset was generated by driving a car on the streets of the city of Turin on October 2, 2017. The two IMUs were installed on a cross-wise aluminium bar mounted on the roof of a car as shown in Figure 6. The direction of motion is indicated by a black arrow on the left. Both of the IMUs’ body frames were aligned in order to match the corresponding axes (*X*, *Y* and *Z*). The Ekinox-D unit was inserted into an aluminium skeleton on which a geodetic GNSS antenna was placed using a plate, centered exactly on the source of the reference system of the inertial sensor (*X*-*Y* center). Then, the antenna was connected to the Ekinox-D. An U-blox M8T GNSS receiver was also included as part of this setup [9], which is the first sensor on the bar in the direction of motion (see Figure 6).

Positions of antennas and sensors were obtained by using a total station and a reflector. Then, they were calculated both in planimetry and altimetry by a small network of distances, following a least-squares approach, reaching a maximum root mean square error of about 2 mm. In this way, it was possible to estimate the lever arms of the system with high accuracy.

The Ekinox-D platform is composed by a dual-frequency and multi-constellation GNSS receiver (GPS: L1, L2; GLONASS: L1, L2, Galileo: E1) and is equipped with a geodetic antenna, while the U-blox is a single-frequency and multi-constellation GNSS receiver (GPS: L1; GLONASS: L1; BeiDou: B1) coupled with a single-frequency patch antenna. These differences between the two GNSS sensors make clear that U-blox is a lower quality sensor. Moreover, according to manufacturers, average horizontal position accuracy is 1.2 m for Ekinox-D [8] and 2.5 m for U-blox [9].

The MPU-600 was configured to operate at a sampling rate of 200 Hz, while Ekinox-D IMU was set to work at its default frequency, 250 Hz. Internal Ekinox-D GNSS receiver and U-blox were both configured to operate at 5 Hz. All sensors were powered by batteries.

The Ekinox-D IMU needs a calibration phase before operation. This previous step allows the Ekinox Kalman filter to calculate the approximate values of the inertial sensors biases that vary at each switch (run-to-run biases). Calibration of this IMU is mandatory in order to get useful information in a post-processing stage. The calibration phase consists of a path of at least ten minutes during which it is performed accelerations and braking at an average speed of at least 30 km/h, while performing some curves in both clockwise and anticlockwise directions. Since this procedure is inconvenient to take place in a normal street, it was run at a parking area at the Politecnico di Torino (Italy), as shown in Figure 7. After the calibration phase, the kinematic trajectory began. It was continuously registered while driving in the streets of the Turin road network.

The kinematic trajectory takes 4.93 km and 21 min (1260 s). It is a typical urban track with heavy leafy foliage and moderate urban canyon effects. The car was driven by open avenues with tall trees. Typically, buildings along the avenues have no more than five floors. This path is denoted as Street stretch. The number of satellites in view during the complete trajectory varies from seven to 11. Additionally, two GNSS-denied paths are forced by removing values of latitude, longitude and altitude at these tracks. Thus, it is simulated that the GNSS receiver stops delivering data during these segments. The GNSS-denied stretches take 35 s each. Figure 8 shows the complete trajectory, where the Street stretch is represented by a yellow line and the GNSS-denied stretches by red lines. Additionally, the blue dot marks the start of the trajectory.

### 3.2. Kinematic Reference Dataset

The reference dataset is built by processing both Ekinox-D measurements, IMU and GNSS data, with tightly coupled integration by using a forward-backward smoothing Kalman filter [13] (Chapter 12) from Inertial Explorer software package (version 8.6) [19]. It comprises attitude and position variables covering both stretches under study (Section 3.1). It has a sampling frequency of 1 Hz.

Furthermore, the GNSS solution is improved by applying the differential GNSS technique using the TORI GNSS permanent station as master station. It is composed by a multi-constellation and multi-frequency receiver and a choke-ring antenna. Its coordinates are estimated with a millimeter level of accuracy in the European Terrestrial Reference System [20].

It is worth mentioning that this reference dataset is the best possible solution that can be obtained by using effectively all available hardware and software resources at hand.

Inertial Explorer provides standard deviations of position and attitude for each point in the reference trajectory. Table 4 shows statistical values of these standard deviations for each navigation variable for the complete trajectory (Section 3.1).

### 3.3. Kalman Filter Tuning

Three loosely coupled INS/GNSS systems are created at a post-processing stage by combining measurements from the two IMUs and the two GNSS receivers, namely, MPU-6000/U-blox (low-cost IMU, low-cost GNSS), MPU-6000/Ekinox (low-cost IMU, mid-range GNSS), and Ekinox/Ekinox (mid-range IMU, mid-range GNSS). The latter is only included in order to compare the performances of the MPU-6000 IMU to a higher grade instrument.

The proposed navigation systems are put into practice by post-processing the kinematic data using NaveGo. NaveGo uses an error-state extended Kalman filter with a conventional 15th-order model approach. Equation (Equation 1) shows the error state to be estimated by the Kalman filter.
(1)$$\begin{array}{cc} \delta \hat{x} & ={\left[\delta {\hat{e}}^{T},\;\delta {\hat{v}}^{nT},\;\delta {\hat{p}}^{nT},\;\delta {\hat{b}}_{g}^{T},\;\delta {\hat{b}}_{a}^{T}\right]}^{T}, \end{array}$$
where $\delta \hat{e}$ is the attitude error vector, $\delta {\hat{v}}^{n}$ is the velocity error vector in the navigation frame, $\delta {\hat{p}}^{n}$ is the position error vector in the navigation frame, and $\delta {\hat{b}}_{g}$ and $\delta {\hat{b}}_{a}$ are the dynamic-biases estimation vectors for gyros and accelerometers, respectively. More information about NaveGo mathematical model and Kalman filter error analysis can be found in [14,21].

Values from Table 1, Table 2 and Table 7 are taken as input values to NaveGo to tune the operation of each INS/GNSS Kalman filter. After several tests, it is concluded that these values do not provide a good navigation performance for the MPU-6000 systems. The reason for this situation is that the MPU-6000 gyroscopes do not follow a Gaussian distribution as shown in Section 2.1, but a pseudo-Gaussian one. Therefore, after several trial-and-error iterations, random walk values from Table 2 for MPU-6000 systems were multiplied by 60, i.e., diagonal values of matrix Q were increased. This way, the Kalman filter is instructed to trust the GNSS measurements more than the inner process model.

### 3.4. Kinematic Results

Table 5 and Table 6 show the root mean squared errors (RMSE) from the three proposed INS/GNSS systems after trial-and-error Kalman filters tuning (Section 3.3). In all cases, no other sensors (e.g., magnetometers) have been used for obtaining the final solutions. These results are obtained considering the fusion process exploiting GNSS and IMU data only.

Additionally, Table 7 exposes the RMSE from the two GNSS-only systems for the complete trajectory, without forcing a GNSS-denied scenario. These values provide an objective assessment of how much improvement in position can be obtained by fusing both GNSS and INS estimations (see Table 5).

Figure 9 shows the attitude errors for the three INS/GNSS systems under analysis for the entire trajectory, including the GNSS-denied paths which are denoted by two black vertical lines. Additionally, Figure 10 exposes the errors in position.

It is possible to see from Table 5 that the Ekinox/Ekinox solution provides a superior performance for attitude for the Street stretch, as expected. However, the MPU-6000/Ekinox setup provides good results in terms of attitude with 3 degrees of dispersion in heading angle. Both systems have similar performances in position, about 2 m in 3D error, which is close to the Ekinox GNSS-only performance (see Table 7), a typical constraint in loosely coupled INS/GNSS systems. On the other hand, inaccuracies in attitude and position from the MPU-6000/U-blox system are the highest among the three INS/GNSS systems. When comparing the two INS/GNSS that include the MPU-6000, a remarkable improvement is achieved in position and heading angle when MPU-6000 is combined with Ekinox GNSS, a higher grade sensor.

In the GNSS-denied stretch, although both Ekinox/Ekinox and MPU-6000/Ekinox systems also show a good track of the heading, the former presents a better roll and pitch accuracy. In terms of positioning accuracy, the three INS/GNSS systems heavily drift from true values, but the two INS/GNSS systems with MPU-6000 show worse position performance when compared to the Ekinox/Ekinox system. In fact, those two systems perform quite similarly except for the heading angle where MPU-6000/Ekinox system is better. The initial estimation of the yaw angle before the start of each GNSS-denied path is the reason for this difference, as shown in Figure 9.

## 4. Concluding Remarks

In this work, an MPU-6000 IMU is evaluated on both static and kinematic scenarios in order to be used as part of a low-cost ground navigation system. A higher performance MEMS IMU, an Ekinox-D, is also included in the experiments just to compare how these two IMUs perform in terms of position and attitude.

Both IMUs are profiled through the Allan variance methodology to quantify stochastic noises. It is observed that MPU-6000 accelerometers have better performances than MPU-6000 gyroscopes. This condition is not optimal for an INS/GNSS system, since attitude errors propagate proportionally to time squared for velocity and time cubed for position. Moreover, MPU-6000 gyroscopes have a lower level of background noise when compared to the Ekinox’s gyroscopes (Table 2), but a higher bias instability. Since the Ekinox/Ekinox system performs better than MPU-6000/Ekinox (see Table 5), this situation exposes that bias instability is a key factor to be compensated in terms of navigation performance. A normality test is performed on the MPU-6000 static data to guarantee that a Kalman filter would be the right filter in the context of an integrated navigation system. These tests reveal that MPU-6000 gyroscopes follow a pseudo-Gaussian distribution. Thus, a trial-and-error tuning of the Kalman filters for MPU-6000 navigation systems have to be conducted.

A kinematic test is done configuring three loosely coupled INS/GNSS systems using two GNSS receivers of different qualities. Especially, performances from the integrated navigation systems are analyzed during three particular paths from an urban trajectory, one with heavy leafy foliage and urban street canyons, and two where the lost of GNSS signal is forced.

Kinematic tests evidence that the MPU-6000 shows very different attitude performances in open-sky trajectories when combined with different GNSS sensors. This situation points out two conclusions. Firstly, MPU-6000 attitude errors seem to be sensitive to GNSS sensor precision and, secondly, heading angle error (yaw angle error) can be about 3 degrees when MPU-6000 is coupled to a mid-grade GNSS receiver. For the GNSS-denied trajectory, the MPU-6000 heading performs also better with a mid-range GNSS.

On the contrary, accuracy in position for MPU-6000 is proportional to the GNSS receiver quality, as expected for loosely coupled INS/GNSS systems. When combined with a mid-range GNSS sensor, 3D position is about 2 m.

In summary, in this work, it is preliminary concluded that the MPU-6000 IMU can be effectively used in mass-market ground navigation systems when coupled to a mid-range GNSS sensor for applications that can tolerate errors of about 2 m for 3D positions and 3 degrees for heading angle.

Finally, it is clear that the performance of ultra low-cost MEMS inertial devices is improving rapidly due to advances in manufacturing techniques. Researchers and engineers in the navigation field should consider this type of sensors for future analysis.

## Figures and Tables

**Figure 1 sensors-19-03865-f001:**
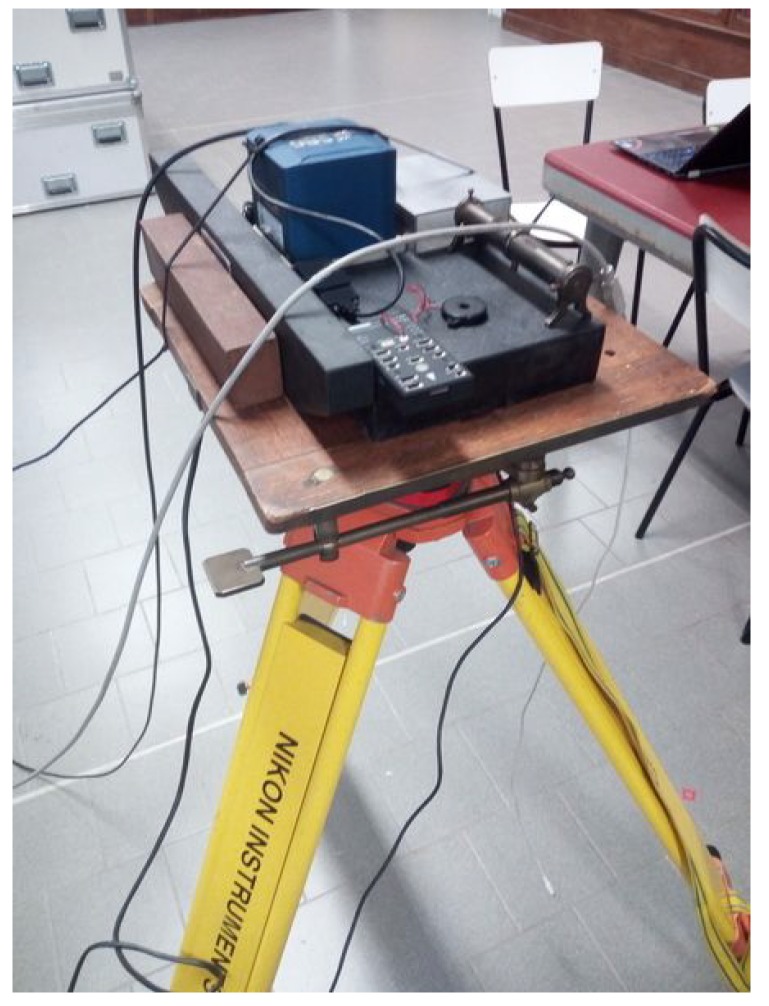
The two IMUs under study mounted on a leveled non-magnetic plate for static logging at DIATI’s laboratories.

**Figure 2 sensors-19-03865-f002:**
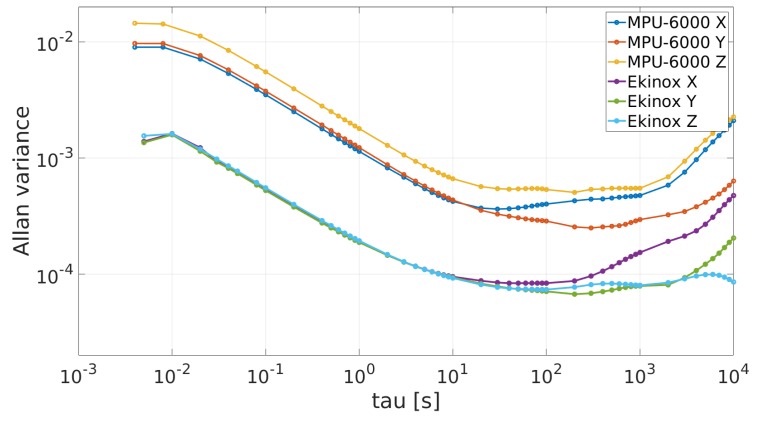
Allan variances plots for accelerometers.

**Figure 3 sensors-19-03865-f003:**
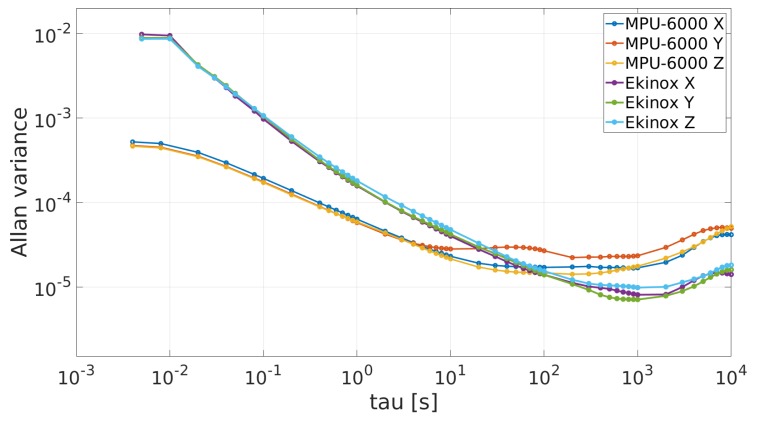
Allan variances plots for gyroscopes.

**Figure 4 sensors-19-03865-f004:**
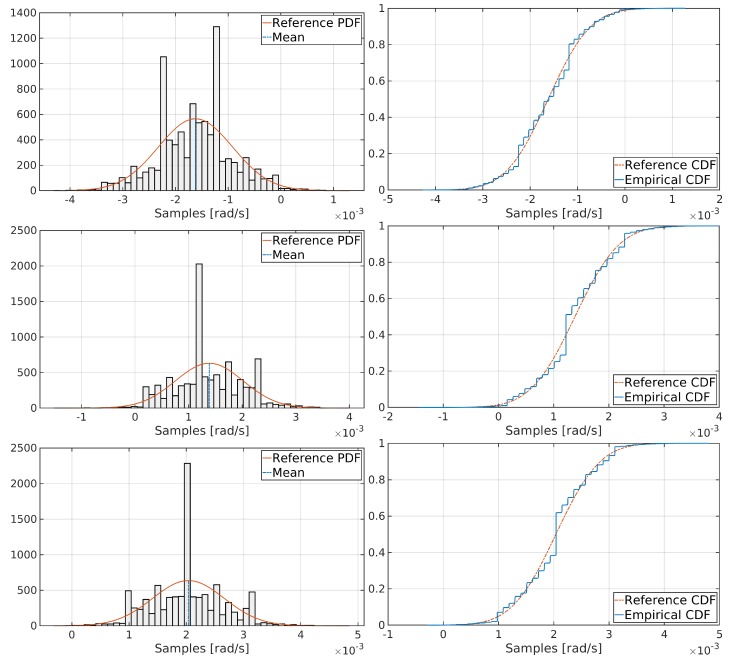
Graphical analysis of normal distribution for X, Y and Z MPU-6000 gyroscopes, respectively.

**Figure 5 sensors-19-03865-f005:**
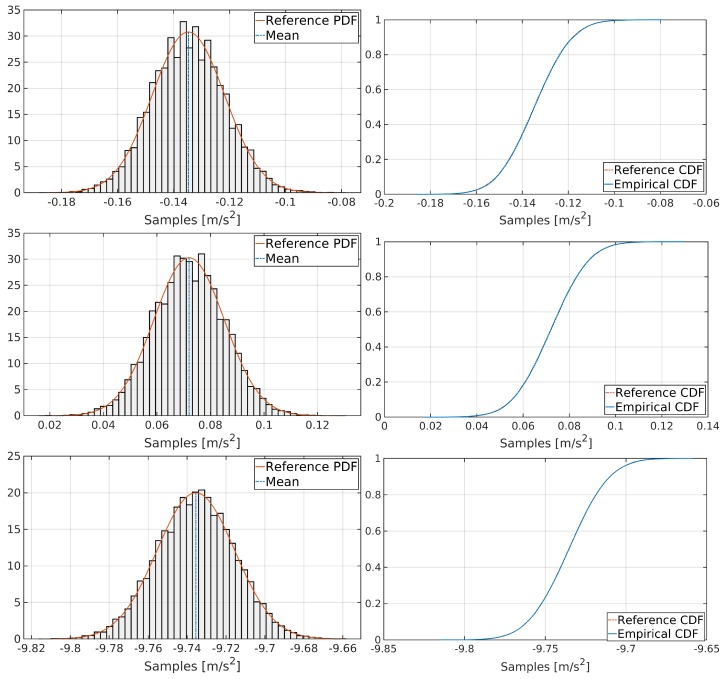
Graphical analysis of normal distribution for X, Y and Z MPU-6000 accelerometers, respectively.

**Figure 6 sensors-19-03865-f006:**
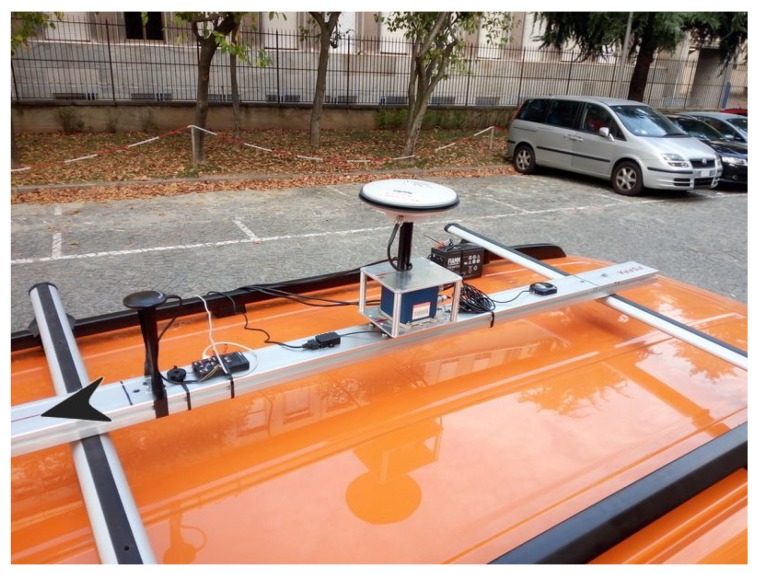
Sensors located on the crosswise bar on top of a vehicle. The black arrow on the left indicates the direction of motion.

**Figure 7 sensors-19-03865-f007:**
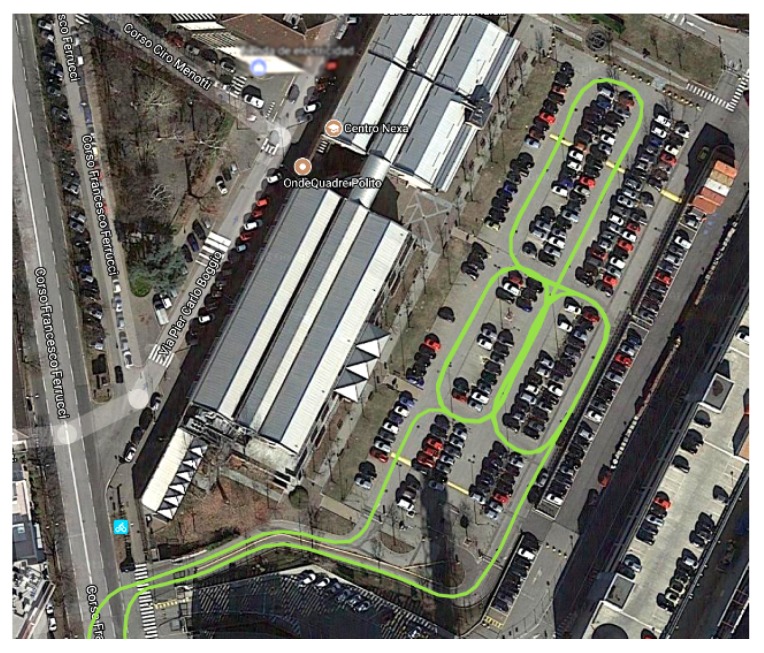
Trajectory during the calibration phase at a parking area at the Politecnico di Torino (image courtesy of Google Maps).

**Figure 8 sensors-19-03865-f008:**
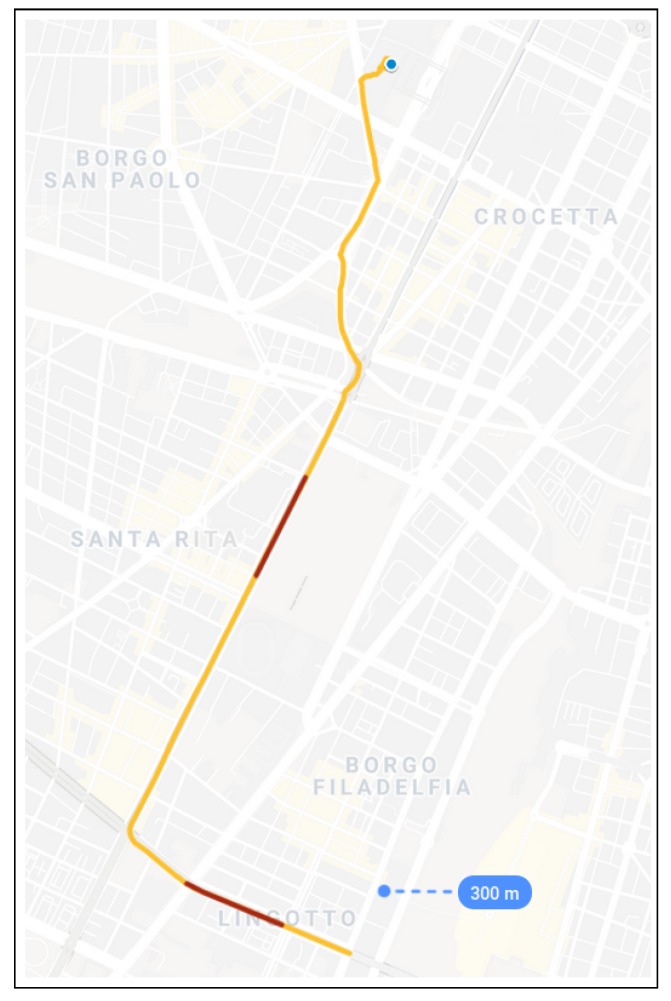
Kinematic trajectory in the streets of Turin. The yellow line depicts the Street stretch, and the red ones, the GNSS-denied stretches. The blue dot shows the starting point (image courtesy of Google Maps).

**Figure 9 sensors-19-03865-f009:**
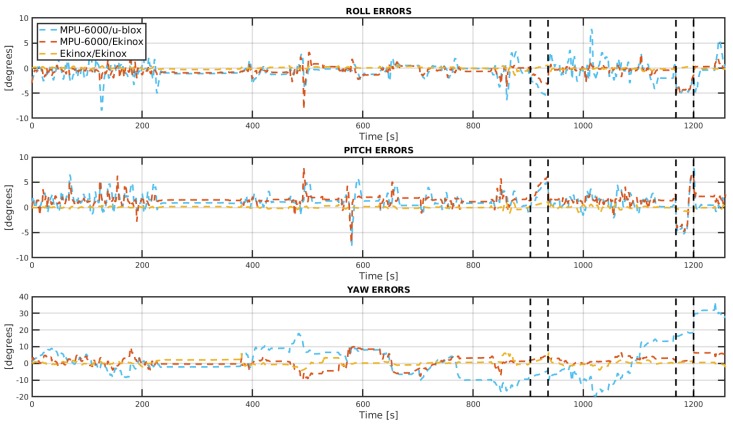
Attitude errors for the complete trajectory. Black vertical lines spot both GNSS-denied stretches.

**Figure 10 sensors-19-03865-f010:**
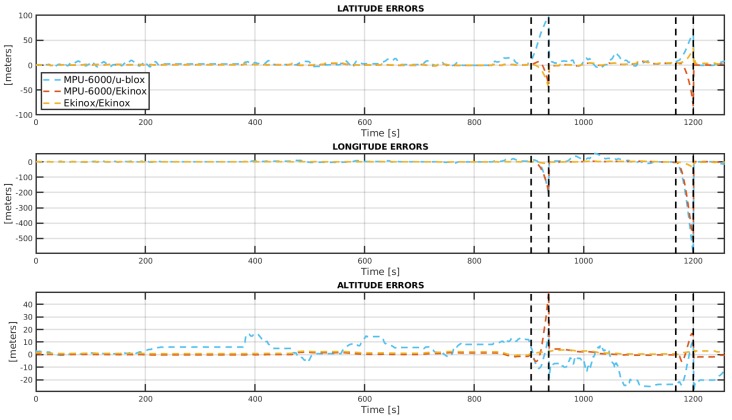
Position errors for the complete trajectory. Black vertical lines spot both GNSS-denied stretches

**Table 1 sensors-19-03865-t001:** Static bias (SB) and standard deviation (σ) from static analysis of the two IMUs under study.

		Static Bias		σ	
		MPU-6000	Ekinox	MPU-6000	Ekinox
Acc X	(m/$s^{2}$)	−1.347 ×$10^{-1}$	1.611 × $10^{-4}$	1.294 × $10^{-2}$	2.293 × $10^{-3}$
Acc Y	(m/$s^{2}$)	7.202 × $10^{-2}$	2.599 × $10^{-2}$	1.324 × $10^{-2}$	2.008 × $10^{-3}$
Acc Z	(m/$s^{2}$)	9.735	9.804	2.012 × $10^{-2}$	2.336 × $10^{-3}$
Gyro X	(rad/s)	−1.616 ×$10^{-3}$	−2.451 ×$10^{-4}$	7.026 × $10^{-4}$	1.023 × $10^{-2}$
Gyro Y	(rad/s)	1.387 × $10^{-3}$	1.230 × $10^{-4}$	6.325 × $10^{-4}$	9.574 × $10^{-3}$
Gyro Z	(rad/s)	2.037 × $10^{-3}$	−2.216 ×$10^{-5}$	6.218 × $10^{-4}$	9.423 × $10^{-3}$

**Table 2 sensors-19-03865-t002:** Noise profile from Allan variance analysis for MPU-6000 and Ekinox IMUs.

	Random Walk			Dynamic Bias		Correlation Time	
	(m$/s^{2}/\sqrt{\mathrm{Hz}}$)(rad/s$/\sqrt{\mathrm{Hz}}$)			(m$/s^{2}$) (rad/s)		(s)	
	MPU-6000		Ekinox	MPU-6000	Ekinox	MPU-6000	Ekinox
Acc X	3.924 × $10^{-3}$	1.156 × $10^{-3}$	1.907 × $10^{-4}$	3.703 × $10^{-4}$	8.393 × $10^{-5}$	30	50
Acc Y	3.924 × $10^{-3}$	1.252 × $10^{-3}$	1.880 × $10^{-4}$	2.501 × $10^{-4}$	6.923 × $10^{-5}$	300	200
Acc Z	3.924 × $10^{-3}$	1.820 × $10^{-3}$	1.939 × $10^{-4}$	5.058 × $10^{-4}$	7.612 × $10^{-5}$	200	100
Gyro X	8.726 × $10^{-5}$	6.625 × $10^{-5}$	1.575 × $10^{-4}$	1.674 × $10^{-5}$	8.098 × $10^{-6}$	900	1000
Gyro Y	8.726 × $10^{-5}$	5.934 × $10^{-5}$	1.603 × $10^{-4}$	2.301 × $10^{-5}$	7.102 × $10^{-6}$	200	1000
Gyro Z	8.726 × $10^{-5}$	6.050 × $10^{-5}$	1.805 × $10^{-4}$	1.462 × $10^{-5}$	9.293 × $10^{-6}$	200	1000

**Table 3 sensors-19-03865-t003:** Normality test based on Anderson-Darling procedure for every inertial sensor from Ekinox and MPU-6000 IMUs. The result *H* is 1 if the test rejects the null hypothesis at the 5% significance level, and 0 otherwise.

	*H*	
	MPU-6000	Ekinox
Acc X	0	0
Acc Y	0	0
Acc Z	0	0
Gyro X	1	0
Gyro Y	1	0
Gyro Z	1	0

**Table 4 sensors-19-03865-t004:** Statistical values of the standard deviations for each navigation variable in the reference dataset for the complete trajectory.

		Mean	Max	Min
Roll	(deg.)	0.0180	0.0459	0.0134
Pitch	(deg.)	0.0174	0.0407	0.0133
Yaw	(deg.)	0.0757	0.1143	0.0492
Latitude	(m)	0.0140	1.4280	0.0040
Longitude	(m)	0.0170	1.7340	0.0040
Altitude	(m)	0.0190	0.1160	0.0070

**Table 5 sensors-19-03865-t005:** RMSE from the three proposed navigation systems for the Street stretch.

		Street		
IMU		MPU-6000	MPU-6000	Ekinox
GNSS		U-blox	Ekinox	Ekinox
Roll	(deg.)	1.604	0.292	0.362
Pitch	(deg.)	1.860	1.752	0.339
Yaw	(deg.)	10.750	3.559	1.839
Lat.	(m)	6.438	1.737	1.256
Lon.	(m)	10.570	1.274	1.084
Alt.	(m)	11.163	1.171	2.631

**Table 6 sensors-19-03865-t006:** RMSE from the three proposed navigation systems for the two GNSS-denied stretches.

		GNSS-Denied 1			GNSS-Denied 2		
IMU		MPU-6000	MPU-6000	Ekinox	MPU-6000	MPU-6000	Ekinox
GNSS		U-blox	Ekinox	Ekinox	U-blox	Ekinox	Ekinox
Roll	(deg.)	4.161	2.265	0.300	4.168	4.173	0.226
Pitch	(deg.)	3.403	4.332	0.703	4.556	4.346	0.715
Yaw	(deg.)	6.530	3.381	2.234	17.743	1.032	0.918
Lat.	(m)	62.131	12.170	17.141	34.489	28.370	14.506
Lon.	(m)	76.106	76.300	6.399	281.99	238.49	16.508
Alt.	(m)	7.733	17.652	1.815	17.070	8.202	1.744

**Table 7 sensors-19-03865-t007:** RMSE from the two GNSS-only solutions.

		Ekinox	U-blox
Lat.	(m)	1.062	5.811
Lon.	(m)	1.063	7.193
Alt.	(m)	3.043	16.177

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
