# Peer review of "Performance Assessment of an Ultra Low-Cost Inertial Measurement Unit for Ground Vehicle Navigation"

_sensors, 2019, doi:10.3390/s19183865_

Round 1
Reviewer 1 Report
This paper presented the performance assessment of an MEMS IMU using Allan variance analysis and road test. The performance assessment is very important nowadays, especially for vehicle navigation which requires high reliability. The results presented in the article is interesting. However, the results are not beyond my expectations. The most important performance of MEMS IMU in vehicle navigation application is the reliability and robustness. To see these, the temperature test, such as bias and scale factor drift over temperature, should be included. The reason why many commercial MEMS sensors cannot be used for car navigation is their limitation on the reliability over temperature, not the performance at certain temperature. Also, it is required for you to test several sensors to check the consistency of the performance. AD normality check is interesting and it can give us very important information. However, for checking the randomness of the sensor output, the noise of power supply also should be analyzed because the severe power noise sometimes may affect the sensor data. And if some frequency analyses such as auto correlation analysis would be conducted, it would be better. In the road test section, the GPS outage period is less than 30 seconds even though you criticized the previous results which considered only short time GPS outages. To verify the performance, some repeated tests or longer simulation trajectories, which contain several GPS outage sections, should be considered. In the experiments, you compared the data with reference composed by smoothing filter and DGNSS. But, I know the Ekinox-D has RTX feature. Why didn’t you use that for the reference? I am not sure why GNSS only results are quite different when you used Ekinox against U-blox. Need more explanation about the reason. Describe road condition, skyscraping condition, etc. Kalman filter results are obvious, and does not tell significant findings.Author Response
Response to Reviewer 1
Dear Reviewer, thank you very much for your invaluable comments. They have let us to improve considerably our manuscript.
This paper presented the performance assessment of an MEMS IMU using Allan variance analysis and road test. The performance assessment is very important nowadays, especially for vehicle navigation which requires high reliability. The results presented in the article is interesting. However, the results are not beyond my expectations.
Point 1: The most important performance of MEMS IMU in vehicle navigation application is the reliability and robustness. To see these, the temperature test, such as bias and scale factor drift over temperature, should be included. The reason why many commercial MEMS sensors cannot be used for car navigation is their limitation on the reliability over temperature, not the performance at certain temperature. Also, it is required for you to test several sensors to check the consistency of the performance.
Response 1: Your comment is quite fair. Temperature is an important aspect to take into account when using MEMS devices. We would like to express that the aim of our current work is to expose preliminary results about the potential use of an ultra low-cost IMU for ground navigation systems. It is our opinion that a deep analysis of the influence of temperature over reliability for ultra low-cost IMUs deserves to be addressed in a complete, independent paper. In fact, we are currently working in modeling the variation of bias versus temperature for MPU-6000 IMU. Later, we will use machine learning algorithms to fit the expected nonlinear relationship that links these two variables.
Point 2: AD normality check is interesting and it can give us very important information. However, for checking the randomness of the sensor output, the noise of power supply also should be analyzed because the severe power noise sometimes may affect the sensor data. And if some frequency analyses such as auto correlation analysis would be conducted, it would be better.
Response 2: All sensors in our experiments were powered by batteries. Thus, we consider that they were not affected by noise from the power supply. We add in section 2 the following statement to our manuscript to clarify this point:
“Both IMUs were powered by batteries, thus they were not affected by electrical noise from the power supply.”
Again in section 3 we make clear that “All sensors were powered by batteries.”
Point 3: In the road test section, the GPS outage period is less than 30 seconds even though you criticized the previous results which considered only short time GPS outages. To verify the performance, some repeated tests or longer simulation trajectories, which contain several GPS outage sections, should be considered.
Response 3: Dear Reviewer, we have extended the trajectory from 1,188 seconds to 1,260 seconds. Additionally, we have added a second GNSS-denied path of 35 seconds. The first GNSS-denied path was set back 100 seconds and increased from 30 to 35 seconds. The reason of the delay was to equally distribute both GNSS-denied paths.
Point 4: In the experiments, you compared the data with reference composed by smoothing filter and DGNSS. But, I know the Ekinox-D has RTX feature. Why didn’t you use that for the reference?
Response 4: Dear Reviewer, we believe that you meant RTK feature for GNSS improvement accuracy. The model that we used, Ekinox-D, as far as we know does not have this feature embedded. This model needs an external device for RTK implementation, which was not available when the kinematic datasef was built.
RTK allows a precision of 1-2 cm on position. Using a combination of Ekinox-D measurements, inertial and GNSS corrected by DGPS, with tightly-coupled integration plus forward and backward (smoothing) processing, we get also an accuracy of 1-2 cm on position, as exposed in Table 4.
Point 5: I am not sure why GNSS only results are quite different when you used Ekinox against U-blox. Need more explanation about the reason.
Response 5: In section 3.1 we have described in more detail the features of both GNSS sensors:
“About some details of the GNSS equipment, the Ekinox-D platform is composed by a dual-frequency and multi-constellation GNSS receiver (GPS: L1, L2; GLONASS: L1, L2, Galileo: E1) and is equipped with a geodetic antenna, while the u-blox is a single-frequency and multi-constellation GNSS receiver (GPS: L1; GLONASS: L1; BeiDou: B1) coupled with a single-frequency patch antenna. These differences between the two GNSS sensors make clear that u-blox is a lower quality sensor. Moreover, according to manufacturers, average horizontal position accuracy is 1.2 m for Ekinox-D [8] and 2.5 m for u-blox [9].”
Point 6: Describe road condition, skyscraping condition, etc.
Response 6: The road condition is described in more detail in section 3.1 as:
“The kinematic trajectory takes 4.93 kilometers and 21 minutes (1260 seconds). It is a typical urban track with heavy leafy foliage and moderate urban canyon effects. The car was driven by open avenues with tall trees. Typically, buildings along the avenues have no more than 5 floors. This path is denoted as Street stretch.”
Point 7: Kalman filter results are obvious, and does not tell significant findings.
Response 7: The aim of section 3.3 (Kalman filter tuning) is to provide awareness of the importance of checking if the Kalman filter is effectively working with data which come from a Gaussian distribution. If not, trial-an-error tests have to be run to tune the R and Q matrices.

Reviewer 2 Report
The paper is very well written and organized. The studied comparative is meaningful and interesting. The methodology is clearly presented and results are presented in an easy to follow and understand way. Results are concluding and point to clear conclusions, and provide significant information in order to the implementation of the studied IMU at the automotive field.
Some minor suggestions may be addressed:
- At line 38, the name of the company figures in a wrong way (InverSense)
- MPU-6000 IMU and Ekinox IMU were logged at 200 Hz and 250 Hz (line 87 and 170-171). Why were they logged at different frequencies? In the case that the frequency for the MPU-6000 were logged at a higher frequency, perhaps the results at the kinematic analysis may exhibit a better behavior?
- Figure 6 is clear but is difficult to know the direction of motion. Labels pointing to the sensors may do easier the identification.
- Results of the MPU-6000 with different GNSS are quite different when the GNSS is denied. This is due to the fact that the initial estimation of the orientation of the IMU is different? Perhaps a discussion about this may be included.
Author Response
Response to Reviewer 2
Dear Reviewer, thank you very much for your invaluable comments. They have let us to improve considerably our manuscript.
The paper is very well written and organized. The studied comparative is meaningful and interesting. The methodology is clearly presented and results are presented in an easy to follow and understand way. Results are concluding and point to clear conclusions, and provide significant information in order to the implementation of the studied IMU at the automotive field.
Some minor suggestions may be addressed:
Point 1: At line 38, the name of the company figures in a wrong way (InverSense)
Response 1: The name of the company was changed to InvenSense.
Point 2: MPU-6000 IMU and Ekinox IMU were logged at 200 Hz and 250 Hz (line 87 and 170-171). Why were they logged at different frequencies? In the case that the frequency for the MPU-6000 were logged at a higher frequency, perhaps the results at the kinematic analysis may exhibit a better behavior?
Response 2: Dear Reviewer, the default operating frequency for Ekinox IMU is 250 Hz and was not changed. On the other hand, MPU-6000 was set to 200 Hz in order to save the MPU-6000 measurements on the SD card without losing any data. We have clarified this point in our manuscript as:
“The Ekinox IMU was logged at 250 Hz, which is the default operating frequency for this sensor. On the other hand, the MPU-6000 IMU was limited to work at 200 Hz to guarantee that MPU-6000 inertial measurements would be saved on the SD card without losing any data.”
Point 3: Figure 6 is clear but is difficult to know the direction of motion. Labels pointing to the sensors may do easier the identification.
Response 3: We have added a black arrow to this figure to indicate the direction of motion of the vehicle.
Point 4: Results of the MPU-6000 with different GNSS are quite different when the GNSS is denied. This is due to the fact that the initial estimation of the orientation of the IMU is different? Perhaps a discussion about this may be included.
Response 4: In this new corrected version, we have added a second GNSS-denied path of 35 second to our work, as requested by Reviewer 1. Additionally, the first GNSS-denied path was set back 100 seconds and increased from 30 to 35 seconds. The reason of the delay was to equally distribute both GNSS-denied paths. In the light of the new RMSE values for both GNSS-denied paths, the two INS/GNSS systems with MPU-6000 perform quite similar except for the heading angle where MPU-6000/Ekinox system is better. As you conveniently suggested, the initial estimation of the yaw angle before the start of each GNSS-denied path is the key factor for this difference as shown in figure 9. We have added this comment in section 3.4.

Reviewer 3 Report
This paper developed an ultra low-cost IMU for ground vehicle navigation. The static data of the output of MPU-600 and Ekinox were sampled, and the static biases, standard deviations, part of Allan variances of accelerometers and gyroscopes were acquired, so the specifications for IMUs were determined. Then test of normality was carried out, a kinematic analysis was made by running the MPU-6000 IMU on the road, the results shows the developed IMU satisfying the demand.
But theoretic innovation of this paper is not stronger, authors can add design method of Kalman filter, including observation analysis, theoretic error analysis into this paper, then contents of this paper will be more integrated.
Some minor errors are as follows:
(1)In table 2, 0.005deg/s/Hz^0.5=8.7266*10^(-5) rad/s/Hz^0.5, this is right; but 400ug/ Hz^0.5 =400*10^(-6)*9.8=3.92*10^(-3)m/s^2/ Hz^0.5, not 3.92*10^(-2)m/s^2/ Hz^0.5, please check carefully, and the second unit may be rad/s/Hz^0.5.
(2)Page 4, line 10. What is “found errors”?
(3) Page 4, the last line, “other” may be “order”.
(4)Page 5,the second line, “which” may be “whose”.
(5) Page 8, “It is denote as”?
(6) Page 11, the tenth from the last line,“Which not is optimal from an INS/GNSS system perspective”?
Author Response
Response to Reviewer 3
Dear Reviewer, thank you very much for your invaluable comments. They have let us to improve considerably our manuscript.
This paper developed an ultra low-cost IMU for ground vehicle navigation. The static data of the output of MPU-600 and Ekinox were sampled, and the static biases, standard deviations, part of Allan variances of accelerometers and gyroscopes were acquired, so the specifications for IMUs were determined. Then test of normality was carried out, a kinematic analysis was made by running the MPU-6000 IMU on the road, the results shows the developed IMU satisfying the demand.
Point 1: But theoretic innovation of this paper is not stronger, authors can add design method of Kalman filter, including observation analysis, theoretic error analysis into this paper, then contents of this paper will be more integrated.
Response 1: Dear Reviewer, we have focused on keep our manuscript neat and clear. Information about the mathematical model of the Kalman filter and algorithms can be found in references [14] and [21]. Nevertheless, we have added in section 3.3:
“More information about NaveGo mathematical model and Kalman filter error analysis can be found at [21] and [14].”
Some minor errors are as follows:
Point 2: In table 2, 0.005deg/s/Hz^0.5=8.7266*10^(-5) rad/s/Hz^0.5, this is right; but 400ug/ Hz^0.5 =400*10^(-6)*9.8=3.92*10^(-3)m/s^2/ Hz^0.5, not 3.92*10^(-2)m/s^2/ Hz^0.5, please check carefully, and the second unit may be rad/s/Hz^0.5.
Response 2: Thank you very much for pointing out these two errors. Values of gyros RW in Table 2 and units of gyros RW were corrected. Conclusions about values from this table were also corrected as well in section 2 as:
“With respect to the MPU-6000’s RW values provided by the manufacturer, RW datasheet values for accelerometers and gyroscopes are close to the values found by applying the AV procedure.”
Point 3: Page 4, line 10. What is “found errors”?
Response 3: “Found errors” was changed for “Errors that come up after AV analysis”.
Point 4: Page 4, the last line, “other” may be “order”.
Response 4: This line was deleted due to the error pointed out in Point 2.
Point 5: Page 5,the second line, “which” may be “whose”.
Response 5: This error was corrected.
Point 6: Page 8, “It is denote as”?
Response 6: The phrase “It is denote as” was replaced by “It is denoted as”.
Point 7: Page 11, the tenth from the last line, “Which not is optimal from an INS/GNSS system perspective”?
Response 7: The phrase “Which not is optimal from an INS/GNSS system perspective” was changed for “This condition not is optimal for an INS/GNSS system”.

Round 2
Reviewer 1 Report
In the revised version, simulations were updated, and some concerns were solved.